# Economic Evaluations of Rehabilitation Interventions: A Scoping Review with Implications for Return to Work Programs

**DOI:** 10.3390/healthcare13101152

**Published:** 2025-05-15

**Authors:** Arie Arizandi Kurnianto, Sándor Kovács, Nagy Ágnes, Prabhat Kumar

**Affiliations:** 1Center for Health Technology Assessment and Pharmacoeconomic Research, Faculty of Pharmacy, University of Pécs, H-7624 Pécs, Hungary; kovacs.sandor@pte.hu (S.K.); nagy.agnes5@pte.hu (N.Á.); 2Institute of Physiology, Medical School, Centre for Neuroscience, Szentágothai Research Centre, University of Pécs, H-7624 Pécs, Hungary; prabhatmbcs@gmail.com; 3Doctoral School of Basic Medicine, Medical School, University of Pécs, H-7624 Pécs, Hungary

**Keywords:** economic evaluation, rehabilitation, return to work, cost-effectiveness

## Abstract

**Background/Objectives**: The use of rehabilitation interventions is critical in addressing health and economic outcomes, including return to work (RTW) facilitation for individuals with disabilities. However, the economic evaluation of these interventions has been found to lack consistency, with limited adherence to reporting standards and little focus on integrated approaches. This scoping review will map the existing evidence on the economic evaluations of rehabilitation interventions and their implications for return to work (RTW) programs. **Methods**: A systematic search of databases, such as PubMed, Scopus, and Web of Science, to identify studies that provided full economic evaluations of rehabilitation interventions related to RTW. Using the PRISMA-ScR framework, 11 studies were ultimately included. Data extraction included the model type, cost-effectiveness models, adherence to CHEERS reporting guidelines, and implications for RTW. **Results**: The majority of studies examined medical or psychological interventions, with little representation of vocational or integrated approaches. ICERs differed greatly between studies based on methodologies and healthcare settings. The reporting of heterogeneity, uncertainty analysis, and societal perspectives were some of the gaps identified from adherence to CHEERS guidelines. **Conclusions**: Economic evaluations show that rehabilitation interventions can be cost-effective for improving RTW outcomes. Future research priorities include interdisciplinary approaches, standardized methodologies, and studies based on LMICs to address global disparities.

## 1. Introduction

Rehabilitation interventions are crucial for enhancing both health and economic outcomes, particularly by facilitating a return to work (RTW) for individuals with disabilities. In fact, recent evidence showed that more comprehensive rehabilitation programs, especially those linked to multidisciplinary approaches and high-intensity interventions, yield positive effects in functional outcomes and quality of life [1]. The rehabilitation interventions not only have a clear clinical benefit, but community-based rehabilitation programs are also more cost-effective than traditional inpatient services, suggesting an opportunity to ensure the optimal allocation of healthcare resources [1].

The economic impact of rehabilitation services has gained prominence in health care decisions. In acute and subacute settings, studies of occupational therapy services reflect variable cost-benefit relationships, with several analyses suggesting that both lower costs, and higher benefits align favorably with higher value [2]. The economic dimension is useful, particularly as healthcare systems are facing pressure to justify the amount of resources allocated without compromising clinical outcomes. Moreover, there is an urgent need for cost-effective rehabilitation solutions, given the increasing incidence of chronic conditions and population aging [3].

In addition, incapable rehabilitation can have serious long-term issues, with research indicating that around 87% of individuals experience chronic symptoms across multiple body systems [4], despite presenting as mild in the first instance. This emphasizes the crucial need for early intervention and suitable rehabilitation programming. In addition, the inclusion of dedicated rehabilitation practitioners in coordination plays a key role in improving the quality of patient outcomes. However, the economic consequences of most such interventions need to be addressed [5]. Vocational and workplace-based occupational rehabilitation programs have demonstrated positive work outcomes in low-back-pain conditions, but cost-effectiveness is less well-evidenced [6].

Furthermore, recent systematic reviews and economic evaluations have shown that it is difficult to assess the value-for-money of rehabilitation interventions. The quality of reporting for economic evaluations of rehabilitation services is inconsistent, which makes it difficult to reach appropriate conclusions [7]. Cost-effectiveness analyses have evolved from a focus on clinical outcomes to recognizing broader societal impacts, such as workplace productivity and quality of life measures [8]. Thus, the evidence indicates the potential for neuropsychological rehabilitation and other specialized neurological stroke care to lead to cost savings and health benefits, but the specific economic effects differ between types of interventions and patient populations [8]. High-quality economic evaluations are needed, conducted with a similar methodology, to ensure robust evidence of the cost-effectiveness of rehabilitation interventions [9]. This variation highlights the need for detailed economic evaluations to guide evidence-based decision making in rehabilitation program design and delivery.

## 2. Materials and Methods

### 2.1. Study Design

In this study, we chose to conduct a scoping review, given the complexity and heterogeneity of the field, to map the breadth and depth of the literature available on the particular topic. Scoping reviews are used to map key concepts, gaps in research, and types of evidence, and are especially relevant for exploring the range of economic evaluations of rehabilitation interventions that facilitate RTW outcomes. Scoping reviews, in contrast to systematic reviews, are broader in focus, allowing for a wide range of study designs and methodologies to be included, provided they address the scope of the questions being asked. This flexibility is crucial to accommodate the diverse range of interventions, outcomes, and cost metrics that these studies employ with respect to economic evaluations in rehabilitation [10,11,12].

This review was conducted in accordance with the PICOS framework for eligibility criteria. The population of interest consisted of patients suffering musculoskeletal disorders undergoing rehabilitation. Interventions included rehabilitation programs specifically designed to improve RTW outcomes (e.g., physical therapy, occupational therapy, or multidisciplinary programs). The comparators were usual care, no intervention, or alternative rehabilitation strategies. Economic outcomes included cost-effectiveness, cost-utility, and cost-benefit analyses and productivity gains. Therefore, to ensure the complete assessment of both costs and consequences, only full economic evaluations were included [10,11]. This framework ensures that the review reflects the studies of relevance to both clinical effectiveness and economic decision-making leverage utility for assigning values to gain in terms of a change in the risk of disease.

Moreover, to assure the integrity and openness of our reports, we complied with the guidance on health economics written for reporting on economic evaluations as described by the Consolidated Health Economic Evaluation Reporting Standards (CHEERS) 2022 checklist [13]. CHEERS is a checklist of recommended items in health economic evaluation reporting that aims to improve and standardize health economic evaluations. CHEERS offers guidance on requisite information for areas such as study perspective, time horizon, cost components, and uncertainty analysis.

### 2.2. Data Sources

The data for this review were obtained from a systematic search of several databases (PubMed, Embase, and Web of Science, among others) to ensure thorough coverage of the literature pertinent to the authors’ question. Terms relating to musculoskeletal disorders, rehabilitation interventions, RTW programs, and economic evaluations were used to develop keywords and search strings. Unpublished or non-peer-reviewed studies that could have shown new information were also thought to be gray literature. The search strategy followed established guidelines (e.g., PRISMA-ScR) for assessing methodological rigor and transparency (see Appendix A) [10,11,12,14].

### 2.3. Data Extraction

We categorized the studies according to characteristics such as the country in which the study was conducted (high-income vs. low-income), the type of intervention (physical therapy or multidisciplinary approaches), the cost element (direct medical costs to indirect productivity losses), and the outcome measures (incremental cost-effectiveness ratios) reported. A narrative synthesis approach was used to summarize the findings across studies, while a thematic analysis was used to identify patterns in the effectiveness and cost-related aspects of interventions. This triangulation ensured that both quantitative information and qualitative themes were explored to gain a comprehensive overview of the economic evaluations of rehabilitation interventions [10,11,15].

## 3. Results

### 3.1. Study Characteristics and Intervention Type

This scoping review is presented according to the Preferred Reporting Items for Systematic Reviews and Meta-Analyses Scoping Reviews (PRISMA-ScR) guidelines [14], shown in Figure 1. From the 867 records found through the databases, 285 were excluded because they were duplicates, leaving 582 records to be screened. At this level, 547 records were excluded based on the screening of titles and abstracts, leaving 35 reports to be downloaded. All reports were downloaded and evaluated for eligibility. In total, 11 studies met the criteria and were included after studies were excluded due to not being a full economic evaluation or focused on health technology interventions or RTW programs or being review papers. The studies were geographically diverse, with representation from Asian countries, European countries, and the United States. The sample sizes ranged from 30 to 10,000 participants (the latter being a hypothetical cohort). The interventions targeted neurological, musculoskeletal, surgical, and psychological disorders and pressure ulcers.

Details of the study characteristics can be found in Table 1. The thematic analysis indicated a predominance of interventions targeting musculoskeletal and neurological disorders with a bias toward individual-level rather than system-level rehabilitation models.

An overview of the included studies, illustrating the geographical diversity and predominance of high-income countries in the studies, is provided in Table 1. This indicates a crucial inadequacy in the representation of low- and middle-income countries (LMICs) where access to rehabilitation services might be a challenge. The trend of publication years indicates that this is an area that is growing in interest over time. This trend highlights the increasing acknowledgment of the requirement for affordable solutions in healthcare systems facing financial limitations. But it also highlights the need for a more contemporary review that considers novel rehabilitation technologies and their economic impact.

Moreover, Table 2 shows the types of interventions carried out across the studies, consisting mainly of medical and psychological interventions. Less widely studied were programs of vocational rehabilitation care, due to the important influence on RTW outcomes. Most studies provided more integrated medical, psychological, and vocational approaches to help people with disabilities with the complex issues they face. More broadly, many studies did not report sufficiently on the intensity and duration of the intervention, which are key factors when judging the possible scalability and sustainability of an approach.

### 3.2. Economic Outcomes

In this study, the information on economic outcomes revealed a significant degree of variation in incremental cost-effectiveness ratios (ICERs) across research studies. Despite the fact that some treatments were found to be cost-effective at willingness-to-pay thresholds (e.g., EUR 10,000–50,000 per QALY), ICERs are high or uncertain for several other interventions due to wide confidence ranges. The variation in ICERs across the studies presented in Table 3 reflects differences in study design, the characteristics of the populations studied, and the nature of the healthcare systems in which the studies were conducted. In many studies, indirect costs such as productivity losses are not consistently reported, making it difficult to assess the full burden on society. It is important that future research focuses on improving comparability across studies through the use of harmonized research methodologies.

To evaluate the compliance of each study with the valuation standards of reporting, the CHEERS criteria were used to assess, particularly as they related to key items such as outcome valuation, cost measurement, and uncertainty characterization, which were each evaluated and recorded on our data extraction sheet [27]. The evaluation of reporting gaps across studies is presented and summarized in Figure 2, which highlights opportunities for improvement, for example, reporting on heterogeneity or the distributional effects of the policy or intervention being evaluated. Our methodology ensured that all economic evaluations were reviewed systematically and that results were interpretable for rehabilitation interventions for evidence-based decision making. Nonetheless, the description outlines why and thereby how the CHEERS checklist was applied and highlights its assurance of sound methods and transparency of reporting for economic evaluations.

Figure 2 shows the reporting status of the perspectives in economic evaluations, which are rated according to CHEERS quality. Among the included studies, five explicitly reported their perspective, two partially reported it, and one adopted a societal perspective. The CHEERS 2022 term “partial reported” is intended to refer to whether the author reported only some aspects of a health economic evaluation rather than in full, That is, the authors provide some but not all of the information for the checklist item. This can mean that the authors discuss or mention the item but fail to provide critical information or did not meet the entire reporting standard [28]. Moreover, perspectives from the healthcare system were underemphasized, reflecting a gap in the understanding of the wider economic impacts beyond direct medical costs. The limited representation of standpoint perspectives underscores the need for future research to incorporate more comprehensive points of view, especially when studying interventions in which productivity and social costs are of vast magnitude. These visualizations unite to show methodological rigor in study quality while at the same time indicating gaps in reporting standards and perspectives across economic evaluations of rehabilitative interventions.

### 3.3. Return to Work Implication

The research articles reviewed in our study provided important insight into how rehabilitation interventions are associated with return to work (RTW) program outcomes (Table 4). For instance, MI, SVAI, and robotics-based interventions reported reductions in lost workdays and improved RTW. The most illustrative was an intervention described by Tingulstad et al. (2023) that reported lost sick days in their sample, i.e., reductions in lost sick days of 7.9 days over six months, with statistically significant reductions in costs [16]. Similarly, Donnelley et al. (2022) reported their prosthesis provision for transfemoral amputees showed an increase in RTW from 10% to 30% in a one-year period and emphasized the value of focused interventions that encourage individuals to reintegrate at work [25]. 

The variation in the intervention type and outcomes of interventions across studies provides variability in standardizing RTW as a metric. Specifically, we note that while several studies reported lost workdays or RTW, others reported indirectly measured outcomes such as increased productivity or reduced healthcare costs. Additionally, few studies became our attention as they either explicitly reported on workplace adaptations or retention in employment, leaving an understanding and use of RTW underreported and underutilized in practice. Continuing this line of research should include descriptive and standardized reporting on metrics for RTW including time to RTW, long-term retention of employment, and consideration of adaptations for work to inform policy and practice.

## 4. Discussion

The purpose of this scoping review is to systematically review the economic evaluations of treatment or rehabilitation programs that are highly likely related to return to work (RTW) programs. It is evident that most of the studies included were from high-income countries, depriving low- and middle-income countries (LMICs) of information, with little access to consistent evidence in the area of rehabilitation informing the RTW process. In fact, studies have shown that low- and middle-income countries (LMICs) experience systemic challenges including insufficient financial and infrastructure resources, which limit the availability and quality of rehabilitation services [29,30]. The stigma and lack of awareness of disabilities and rehabilitation services limits access to services. It is not easy to overcome obstacles to access rehabilitation services in LMICs; the implementation of community-based rehabilitation (CBR) programs may reduce the barriers to rehabilitation by delivering rehabilitation and disability support services directly to communities, being grounded in local resources [29]. The geographic imbalance highlights the need for more health economics research to mitigate inequalities in rehabilitation services access, economic evaluations, and the RTW process in LMICs, in the context of rehabilitation with sustainable health services. Among the 11 studies included, 8 (82%) were published from 2020 onwards, indicating that economic evaluation in rehabilitation has recently become a priority. Most studies (n = 9) were published between 2020 and 2023, indicating a surge in interest in cost-effective rehabilitation approaches following the COVID-19 pandemic. The increasing number of studies indicates that researchers are striving to synthesize the growing knowledge about new rehabilitation technologies and understand how these technologies can be integrated, where possible, into the current healthcare framework. Although our sample size is small, the summary of the most recent studies included is consistent with recent samples in the field of health economics, where cost-effectiveness analysis is gaining considerable attention.

Most of the studies included had medical and psychological interventions, with fewer studies focusing on vocational rehabilitation programs, even though they have the best opportunity to improve RTW outcomes. Though we noted attention to medical and psychological treatments, vocational rehabilitation programs were not widely described, which is consistent with the previous study that also pointed out how evidence was scant on vocational interventions notwithstanding their importance for RTW outcomes [31]. Integrated models of RTW program that combined medical, psychological, and vocational approaches were also rare, but there is some indication they can help to address the multidimensional nature of the challenges of people with disabilities [21]. In addition, some of the studies reported on intensity and duration with little detail, which minimally affects how we assess scalability and sustainability. These limitations highlight the need for standardized reporting frameworks to facilitate the comparison of similar studies and to help guide decision making based on evidence.

In addition to issues of underrepresentation in the literature, vocational rehabilitation must be viewed and understood as a wider continuum of care that does not strictly adhere to clinical programming. Furthermore, the rehabilitation and readaptation process, along with workplace reintegration, interactions with an employer, and the coordination of health service systems with labor markets, is affixed in adaptable positions [32]. This is important since integrated approaches to services highlight opportunities for prevention and continuity of care based on long-term functional and occupational reintegration—crucial elements when determining integrated RTW policies that address both individual dignity and system efficiency.

Studies were heterogeneous with no consistent pattern in incremental cost-effectiveness ratios (ICERs), due to variabilities in study designs, population characteristics, and healthcare system trade-offs. Some interventions were cost-effective within acceptable willingness-to-pay levels (e.g., EUR 10,000–50,000 per QALY), whereas others had relatively high incremental cost-effectiveness ratios (ICERs) or uncertainty due to high confidence intervals. The variation in ICERs was also great in our review, similar to the findings of a study that attributed the high variability in ICERs to differences in the context of healthcare systems and methodologies [33]. Future economic evaluations could include more long-term outcomes and cost-benefit perspectives that go beyond the healthcare sector. Future research should include the development and evaluation of non-specialist training programs and the incorporation of rehabilitation concepts throughout health workforce education [34]. Furthermore, there were inconsistently reported indirect costs, like productivity losses, preventing a complete assessment of societal impacts. The harmonization of methodologies should be the focus of future research to improve comparability across studies and provide robust evidence for policymakers.

Studies showed variation in ICERs, which highlights the ways in which cost-effective methods and cost-effectiveness contexts diverge from each other, such as differences in population characteristics, cost categories, and perspective. This makes synthesis difficult, and finding opportunities to transfer findings is limited. Moving forward, to facilitate the process of comparing studies, systematic cost-effectiveness studies should share a common meta-framework for economic evaluation and standardized modes for reporting economic evidence (e.g., cost categories, time horizon, and societal versus healthcare perspectives). If studies were to standardize these aspects, that would help move toward the indirect comparison and meta-synthesis of studies in systematic reviews in the future.

The employment factors, such as a decrease in sickness absence days, noted with SVAI interventions parallel significant RTW improvements in the literature following similar tailored vocational programs [35]. The rehabilitation interventions can produce positive effects on RTW outcomes (e.g., number of days lost from work, time taken to return to the labor market). For instance, one of the included studies in our review found a reduction of 7.9 sickness absence days with SVAI [16], while another study reported that RTW rates post-prosthesis provision were 20% higher for transfemoral amputees [25]. However, the standardization of RTW measures is difficult across studies due to variability in interventions and measurement outcomes. Very few studies focused on workplace adaptations or long-term employment retention that would allow for a richer understanding of sustainable RTW outcomes. Future research should develop standardized RTW outcomes under consistent conditions (time taken to return to work, long-term work retention, workplace adaptations) that will help guide evidence-based policy and practice and identify barriers to returning to work.

The significance of health policy planning considerations may be expressed as clear goals and guidelines for an RTW program. Clearly stated goals and guidelines are needed to implement RTW policies and programs effectively. Policies must be evidence-based and include practice guidelines to ensure consistent practice and clear execution of RTW plans [36]. Additionally, training requires ensuring that healthcare providers have the competencies required to implement optimal RTW plans. This includes recognizing how to write restrictions that are clear and easily implemented by both employees and employers. Once again, ongoing education and support for clinicians can enable improved communication and collaboration with patients and employers [37].

In addition, important deficiencies in the reporting standards among the included studies in our review have been identified using the CHEERS checklist, especially in defining heterogeneity, distributional effects, and uncertainty analysis. Most studies met the minimum reporting standards for cost components and perspective but fewer reported on broader societal impacts or engaged stakeholders in their study design. Such gaps diminish the visibility of, and ability to apply, study results to inform decision making. First, there is a need to improve adherence to CHEERS in order to improve the quality and consistency of economic evaluations. Moreover, our analysis identified gaps in reporting on heterogeneity and uncertainty, aspects of relevance for policy decisions. Many economic evaluations underreported on patient heterogeneity, including failing to discuss whether patient populations differed, which affects generalizability and applicability. Further, studies do not report the full range of uncertainty analyses that are fundamental to good decision making [38,39].

### Strengths and Limitations

This scoping review provides an overview of the existing economic evaluations of rehabilitation interventions and their relevance for return to work (RTW) programs. It followed the international standards of methods such as PRISMA-ScR and CHEERS that ensured transparency and reproducibility. The review integrates findings from different types of interventions—medical, psychological, and vocational—to highlight how rehabilitation programs are fundamentally complex. Also, studies have been conducted from diverse healthcare systems, showcasing differences in cost-effectiveness thresholds, perspectives, and economic evaluation practices internationally. In addition, the identification of gaps in reporting standards and RTW metrics—specifically with respect to RTW measurement—will help to improve the methodological rigor in future research.

However, this review has a number of limitations. First, the included studies were mostly implemented in high-income countries, which potentially limits the applicability of results to LMICs where healthcare access and resources vary greatly. Second, many studies did not report in detail the intensity, duration, and scalability of outcomes resulting from interventions, and all of these factors are key considerations when assessing potential for sustainability long term. Third, direct comparisons are difficult because of differences in methodologies among studies, for instance, differences in ICER calculations and time horizons. In conclusion, although RTW outcomes were reported, studies focusing on long-term workplace accommodations or sustainable employment retention in the workplace were absent, preventing a complete assessment of RTW outcomes.

## 5. Conclusions

This review of the field indicates the need for economic evaluations to better understand the value of rehabilitation interventions especially within RTW programs. The evidence indicates much is known about medical and psychological interventions but less is known about multi-faceted vocational or integrated rehabilitation approaches, which have the potential to address the complex challenges of persons with disabilities. Rehabilitation has the potential to improve RTW outcomes by reducing lost workdays and incorporating a return to work into the workforce. It was observed that the considerable variation in metrics and a lack of focused consideration of workplace accommodations clearly indicates a new way forward to improve the delivery of rehabilitation services. To fill these gaps and foster methodological uniformity, we recommend that an international working group be established with the aim of developing a standard framework for conducting and reporting economic evaluations in rehabilitation with a RTW focus. Such collaboration will be key to ensuring evidence-based and equitable rehabilitation methods in different health systems.

## Figures and Tables

**Figure 1 healthcare-13-01152-f001:**
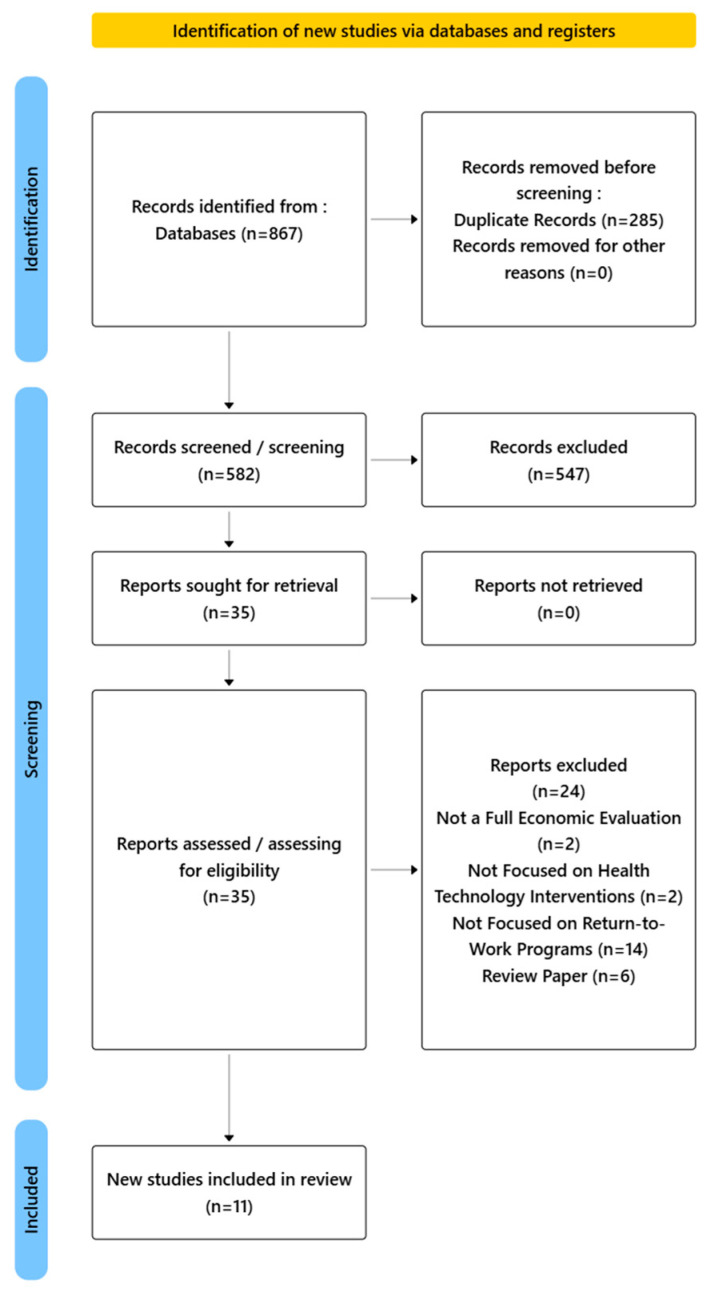
Study framework.

**Figure 2 healthcare-13-01152-f002:**
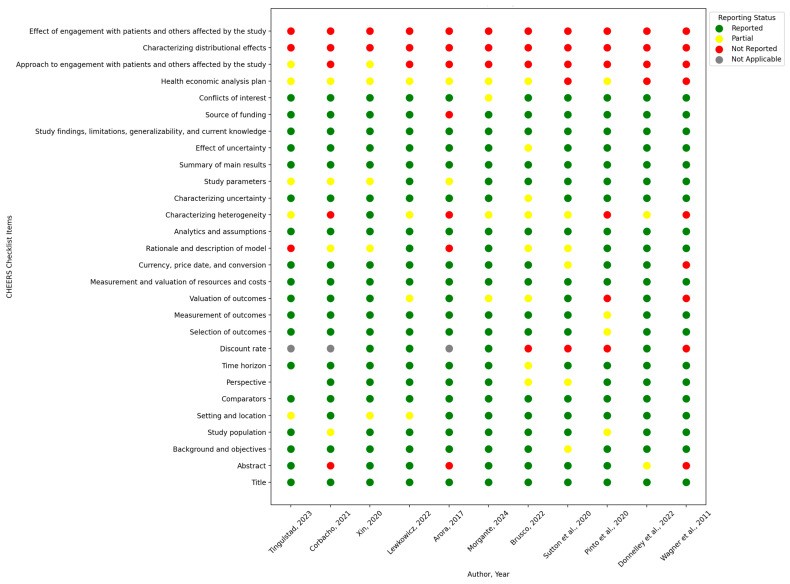
CHEERS validation table: reporting status across studies [16,17,18,19,20,21,22,23,24,25,26].

**Table 1 healthcare-13-01152-t001:** Characteristics of the included studies.

Author, Year	Country	Type of Disease	Type of Disability	Age	Number of Patients
Tingulstad, 2023 [16]	Norway	Musculoskeletal disorder	Musculoskeletal disorders	Median 49 years (range 24-66)	514
Corbacho, 2021 [17]	United Kingdom	Frozen shoulder	Musculoskeletal disorders	Not Reported	503
Xin, 2020 [18]	United Kingdom	Parkinson’s disease	Neurological disorders	Mean age 71 (SD 7.7) in intervention group, 73 (SD 7.7) in control group	474
Lewkowicz, 2022 [19]	Germany	Low-back pain	Musculoskeletal disorders; Psychological Disorders	41 years (mean)	10,000
Arora, 2017 [20]	India; Bangladesh	Pressure ulcers in spinal cord injury	Neurological disorders	35 (SD 11) in intervention group, 36 (SD 12) in control group	115
Morgante, 2024 [21]	Norway	Musculoskeletal and psychological disorders	Musculoskeletal disorders; psychological disorders	40 years (mean age)	166
Brusco, 2022 [22]	Australia	Stroke	Neurological	Not Reported	30
Sutton et al., 2020 [23]	United States	Spinal cord injury	Neurological disorders	51.0 ± 10.1 years (PrOMOTE group), 49.8 ± 9.8 years (TAU group)	213
Pinto et al., 2020 [24]	United States	Spinal cord injury (SCI)	Neurological disorders	Mean age 43 years	Not Reported
Donnelley et al., 2022 [25]	Tanzania	Transfemoral amputation	Physical disability	Mean 45.9 years (SD 17.6)	38
Wagner et al., 2011 [26]	United States	Stroke	Neurological disorders	Not Reported	127

**Table 2 healthcare-13-01152-t002:** Description of intervention type.

Author, Year	Intervention	Comparator	Program Details	Type of Rehabilitation Intervention
Tingulstad, 2023 [16]	Motivational interviewing (MI) and stratified care	Usual case management (UC)	Motivational interviewing sessions and vocational support	Medical (physiotherapy); vocational (stratified vocational advice intervention—SVAI); psychological (motivational interviewing—MI)
Corbacho, 2021 [17]	Manipulation under anesthesia (MUA), arthroscopic surgery	Standardized physiotherapy programs	Standardized physiotherapy programs, intra-articular injections	Medical: Early structured physiotherapy (ESP) with steroid injection; Vocational: not reported; Psychological: not reported
Xin, 2020 [18]	PDSAFE intervention	Usual care	Personalized physiotherapy program targeting balance and mobility	Medical: Personalized physiotherapy program
Lewkowicz, 2022 [19]	Digital therapeutic care app	Treatment as usual	Digital therapeutic care app with video-based guidance and feedback	Medical (digital therapeutic care app with video-based exercises and educational material); Psychological (decision-support interventions)
Arora, 2017 [20]	Telephone-based support	Usual care	Weekly telephone support reinforcing self-help strategies	Medical (education on wound care, diet, equipment use); Psychological (stress management); Vocational (self-help strategies)
Morgante, 2024 [21]	O-ACT (outpatient acceptance and commitment therapy)	Usual care	O-ACT: 6-week program with weekly sessions, homework assignments, and follow-up	O-ACT: Outpatient acceptance and commitment therapy (psychological); I-MORE: Inpatient multimodal occupational rehabilitation (medical, psychological, vocational)
Brusco, 2022 [22]	Robotics-based therapy (RBT)	Standardized outpatient therapy	Robotics-based therapy with standardized protocols for rehabilitation	Medical (robotics-based therapy for upper-limb rehabilitation); Conventional therapy (physiotherapy and occupational therapy)
Sutton et al., 2020 [23]	Supported employment program based on individual placement	Usual care	Individual placement and support model focusing on job placement and support	Vocational (individual placement and support employment program)
Pinto et al., 2020 [24]	Robotic exoskeleton over-ground training	Standard rehabilitation	Robotic exoskeleton training for locomotor rehabilitation	Robotic exoskeleton training (RT-exo), body-weight-supported treadmill training (BWSTT), over-ground training (OGT), stationary robotic systems (treadmill-based robotic gait orthoses)
Donnelley et al., 2022 [25]	Prosthesis provision	No prosthesis	Provision of modular endoskeletal transfemoral prosthesis with follow-up care	Medical: Prosthesis provision and gait training
Wagner et al., 2011 [26]	Robot-assisted therapy	Intensive comparison therapy	Robot-assisted therapy and intensive comparison to standard rehabilitation methods	Medical: Robot-assisted therapy; Intensive comparison therapy; Usual care

**Table 3 healthcare-13-01152-t003:** Economic evaluation among the included studies.

Author, Year	Economic Outcomes	Time Horizon	Threshold	ICER
Tingulstad, 2023 [16]	Cost-Effectiveness Analysis (CEA)	6 months	NOK 275,000 (EUR 27,500/USD 35,628) per QALY	ICER for MI: EUR 1,756,221 per QALY; ICER for SVAI: EUR 1,553,061 per QALY
Corbacho, 2021 [17]	Cost Utility Analysis (CUA)	1 year	GBP 20,000/QALY	GBP 6984 per additional QALY for MUA compared to ESP
Xin, 2020 [18]	Cost Utility Analysis (CUA)	6 months; 12 months in sensitivity analysis	GBP 30,000/QALY	GBP 120,659 per QALY
Lewkowicz, 2022 [19]	Cost-Effectiveness Analysis (CEA)	3 years	EUR 10,000/QALY and EUR 20,000/QALY	EUR 5486 per QALY
Arora, 2017 [20]	Cost-Effectiveness Analysis (CEA)	12 weeks	Less than three times GDP per capita	INR 2306 (USD 130) per cm^2^ reduction; INR 44,915 (USD 2523) per QALY gained
Morgante, 2024 [21]	Cost-Effectiveness Analysis (CEA)	25 years	USD 50,000 per QALY	USD 356,447 per QALY gained (healthcare perspective); I-MORE dominated O-ACT (societal perspective)
Brusco, 2022 [22]	Cost-Effectiveness Analysis (CEA)	6 months (6 program cycles)	Not Reported	Not Reported
Sutton et al., 2020 [23]	Cost-Effectiveness Analysis (CEA)	2 years	USD 50,000/QALY; USD 100,000/QALY	Not Reported
Pinto et al., 2020 [24]	Budget Impact Analysis (BIA)	1 year	Not Reported	Not Reported
Donnelley et al., 2022 [25]	Cost-Effectiveness Analysis (CEA)	1 year (study duration); Lifetime (modeled)	USD 1080 (conservative); USD 3140 (PPP-adjusted)	USD 242/QALY (payer, no replacement); USD 390/QALY (payer, with replacement); Dominated (societal perspective)
Wagner et al., 2011 [26]	Cost-Effectiveness Analysis (CEA)	36 weeks	Not reported	Wide confidence region (-USD 450,255 to USD 393,356); uncertainty remains about cost-effectiveness

**Table 4 healthcare-13-01152-t004:** Return to work (RTW) implications.

Author, Year	RTW Implication
Tingulstad, 2023 [16]	Reduction in sickness absence days: 5.1 days (UC + MI) and 7.9 days (UC + SVAI) over six months
Corbacho, 2021 [17]	Median lost workdays: 14 days (ESP), 56.5 days (MUA), 71.5 days (ACR)
Xin, 2020 [18]	Not Reported
Lewkowicz, 2022 [19]	8 workdays lost per cycle
Arora, 2017 [20]	8 workdays lost per cycle
Morgante, 2024 [21]	Workdays lost: O-ACT (GB: 21 days, DB: 24 days); I-MORE (GB: 20 days, DB: 18 days); I-MORE presents a faster RTW than O-ACT
Brusco, 2022 [22]	Not reported
Sutton et al., 2020 [23]	Not reported
Pinto et al., 2020 [24]	Cost savings from RT-exo adoption ranged from USD 1114 to USD 4784 annually across facilities
Donnelley et al., 2022 [25]	Return to work increased from 10% to 30% after 1 year; 20% increase attributed to prosthesis provision
Wagner et al., 2011 [26]	Not Reported

## Data Availability

The data that support the findings of this study are available upon reasonable request from the corresponding authors.

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
