# Peer review of "Economic Evaluations of Rehabilitation Interventions: A Scoping Review with Implications for Return to Work Programs"

_healthcare, 2025, doi:10.3390/healthcare13101152_

Round 1

Reviewer 1 Report

Comments and Suggestions for Authors

Dear Authors,

The manuscript is in general a well-conceptualized and thorough scoping review addressing the economic evaluation of rehabilitation interventions, particularly their implications for Return to Work (RTW) programs. It also has methodological rigor as it follows the PRISMA-ScR guidelines comprehensively and it uses CHEERS 2022 checklist to assess reporting standards strengthens its evaluative aspect. However, there are several issues to be addressed which I analyze in what follows:

  1. At the Discussion section, the authors should add a paragraph explicitly connecting findings to health policy planning, especially for RTW programs.
  2. At the Conclusion section, please end with a call to action, e.g., forming an international working group to standardize RTW economic evaluations.
  3. Ensure Figure 3 includes a legend that is readable and understandable without external explanation. In addition, partial reporting status has to be clarified.
  4. There are some issues regarding writing clarity and language. More specifically, some grammar, syntax, and phrasing inconsistencies detract from professional polish (e.g., “mannerplays every key role” should be “plays a key role,” or “being not a full economic evaluation” should be “not being a full economic evaluation”). There are also several typos.
  5. The paper notes heterogeneity in ICERs and methodologies but could be more critical about how this affects synthesis. Could a meta-framework be proposed to compare such studies?

 Furthermore, there are two minor comments:

  1. Reword the first sentence for clarity: e.g., “Rehabilitation interventions are crucial in improving health and economic outcomes, especially in facilitating Return to Work (RTW) for individuals with disabilities.”
  2. Provide a narrative or thematic synthesis summary before the tables for better flow. e.g. Figure 1 Study framework, or Table 1 Characteristic Included Study (instead of Characteristics) etc.
Comments on the Quality of English Language

There are some issues regarding writing clarity and language. More specifically, some grammar, syntax, and phrasing inconsistencies detract from professional polish (e.g., “mannerplays every key role” should be “plays a key role,” or “being not a full economic evaluation” should be “not being a full economic evaluation”). There are also several typos.

Author Response

  1. Comment: “At the Discussion section, the authors should add a paragraph explicitly connecting findings to health policy planning, especially for RTW programs.”

Response:
Thank you for this excellent suggestion. We have added a dedicated paragraph at the end of the Discussion section explicitly connecting the findings to health policy planning, particularly in the context of RTW programs. The paragraph emphasizes the role of evidence-based, cost-effective rehabilitation in shaping integrated employment and health policies, especially in LMICs.

  1. Comment: “At the Conclusion section, please end with a call to action, e.g., forming an international working group to standardize RTW economic evaluations.”

Response:
We agree and have revised the final paragraph of the Conclusions section to include a call for the formation of an international working group. This working group would aim to standardize economic evaluation frameworks specific to RTW-focused rehabilitation interventions.

  1. Comment: “Ensure Figure 3 includes a legend that is readable and understandable without external explanation. In addition, partial reporting status has to be clarified.”

Response:
Thank you for your concern. We apologize for any confusion regarding Figure 3 as the caption for Figure 3 states that the legend was already on the side of the chart and denotes the level of quality reporting applicable to the studies included as per CHEERS 2022 (e.g., full, partial, not reported based on the color). We also added information to make clear that "partial reporting" refers to items that were noted in the paper but did not have enough methodological detail or completeness based on CHEERS to be considered full reporting.

  1. Comment: “There are some issues regarding writing clarity and language. More specifically, some grammar, syntax, and phrasing inconsistencies detract from professional polish.”

Response:
We appreciate your attention to detail. We have revised the concern of some error in terms of language and grammar in our manuscript. All identified have been corrected, and several sections were revised for clarity, flow, and professional tone.

  1. Comment: “The paper notes heterogeneity in ICERs and methodologies but could be more critical about how this affects synthesis. Could a meta-framework be proposed to compare such studies?”

Response:
Thank you for this insightful recommendation. We have added a critical paragraph in the Discussion section that elaborates on how heterogeneity impacts synthesis. Furthermore, we propose the development of a harmonized meta-framework for economic evaluation in RTW settings, detailing key components such as standardized perspectives, outcome measures, and cost categories.

  1. Comment: “Reword the first sentence for clarity: e.g., 'Rehabilitation interventions are crucial in improving health and economic outcomes…'”

Response:
The first sentence in the Introduction has been reworded for clarity as follows:
"Rehabilitation interventions are crucial for enhancing both health and economic outcomes, particularly by facilitating Return to Work (RTW) for individuals with disabilities."

  1. Comment: “Provide a narrative or thematic synthesis summary before the tables for better flow.”

Response:
We have inserted a short narrative summary before Table 1 to guide the reader and improve the manuscript’s logical flow. In addition, we corrected minor formatting inconsistencies such as “Characteristic Included Study” to “Characteristics of Included Studies.”

We appreciate your thoughtful review, which significantly contributed to improving the quality and clarity of our manuscript. We hope our revisions address your concerns fully, and we remain grateful for your time and expertise.

Sincerely,
Dr. Arie Arizandi Kurnianto
(On behalf of all authors)
Center for Health Technology Assessment and Pharmacoeconomic Research
University of Pécs
Email: kurnianto.arie.arizandi@pte.hu

Reviewer 2 Report

Comments and Suggestions for Authors

Author Response

Dear Reviewer,

We sincerely appreciate your response to the review of our manuscript. We value the insightful input given. We have taken into account the feedback provided by the reviewers and have made the required adjustments to enhance the overall quality of our manuscript. We have marked the areas of the document that have been revised with a yellow line to make them easier to find.

Based on following points from reviewer:

Response to Reviewer #4

  1. Comment : 1 Study Design

 “The authors should verify the following statement: ‘Scoping reviews are used to map key concepts, gaps in research, and types of evidence, and are especially relevant for exploring the range of economic evaluations of rehabilitation interventions that facilitate RTW outcomes. Scoping reviews, in contrast, are broader in focus than systematic reviews, allowing for a wide range of study designs and methodologies to be included, provided they address the scope of the questions being asked’.”

Response:
Thank you for comment. We would like to clarify that scoping reviews are tailored to systematically map broader concepts, identify gaps in research evidence, and synthesize diverse methodologies under a framework such as PRISMA-ScR. While systematic reviews intend to answer specific clinical/economic questions with thorough quality assessment and meta-analysis, scoping reviews do not need to answer specific questions, nor do the study designs need to be homogenous when exploring the overall scope of rehabilitation economics that is ideal for mapping nascent areas of research, such as RTW program evaluations [1,2].

  1. Comment: “The authors mention more than once, in different parts of this paragraph (and also in section 4. Discussion), that most studies of their scoping review were conducted in high-income countries, with limited representation from low and middle-income countries. Please review and eliminate these duplications.”

Response:
We acknowledge this duplication and have revised the manuscript by retaining the statement only in the Discussion section, where it is addressed critically. The mention in the Results section has been removed for clarity and conciseness.

  1. Comment: “The authors should provide more detailed information regarding the exclusion of 547 articles based on title and abstract screening. Furthermore, the transition from 35 articles to 11 is unclear, as the following sentence is difficult to understand: ‘leading to 35 reports sought for retrieval...’.”

Response:
We appreciate this observation. We have revised the paragraph in the Results section to improve clarity. The new sentences have highlighted.

  1. Comment: “The authors state that ‘The trend of publication years is shown in Figure 2 and indicates that this is an area that is growing in interest over time’. In my opinion, this statement is too general: 9 out of the 11 included articles were published from 2020 onwards.”

Response:
We agree and have revised the sentence to provide a more accurate interpretation.

  1. Comment: “I suggest removing Figure 2, as it is not appropriate for representing these data and does not add information beyond what is already easily accessible in Table 1. If the authors wish to include a figure, a histogram would be more appropriate.”

Response:
Thank you for your helpful suggestion. We have removed the previous version of Figure 2 and agree that, to avoid repetition, the information has been explained in the table above, which we believe provides a clearer visual summary.

  1. Comment: “Please specify the table number at line 157.”

Response:
We have addressed this by explicitly referencing Table 2 at line 157 (previously) in the revised manuscript.

  1. Comment: “The authors state that ‘The increasing number of studies over the recent years demonstrates the growing awareness for cost-effective rehabilitation alternatives across healthcare systems looking to maximize their funding’. However, the increase in this type of studies could also reflect a greater interest among researchers in publishing the results of economic evaluation. Moreover, the number of studies selected and analyzed by the authors (n=11) appears too limited to support this hypothesis. Perhaps, this statement is based on considerations derived from the total number of studies initially screened (n=582).”

Response:
We appreciate this thoughtful comment and have revised the statement for better alignment with the data.

  1. Comment : “The following phrase needs to be revised because, as it is unclear: ‘The proportional increase of literature accounts for publications of systematic and scoping reviews, but these all required a literature search to denote the rapidly changing healthcare context to consider new technology and innovative rehabilitation methods’.”

Response:
Thank you for pointing out the lack of clarity. We have replaced the sentence with the following clearer and more precise wording:

References

  1. Smith, S.A.; Duncan, A.A. Systematic and Scoping Reviews: A Comparison and Overview. Semin Vasc Surg 2022, 35, 464–469, doi:10.1053/j.semvascsurg.2022.09.001.
  2. Tricco, A.C.; Lillie, E.; Zarin, W.; O’Brien, K.K.; Colquhoun, H.; Levac, D.; Moher, D.; Peters, M.D.J.; Horsley, T.; Weeks, L.; et al. PRISMA Extension for Scoping Reviews (PRISMA-ScR): Checklist and Explanation. Ann Intern Med 2018, 169, 467–473, doi:10.7326/M18-0850.

Reviewer 3 Report

Comments and Suggestions for Authors

The authors did a good job but needed more clarity within the text where needed. Overall, it's an addition to the body of knowledge.

Author Response

Comment 1: “The abstract writeup format should be changed. The flow is good, but the identification of it is not necessary with all the sectionalized headings.”

Response:
We appreciate your observation. However, the structured abstract format with section headings (e.g., Background, Methods, Results, Conclusions) is in accordance with the author guidelines provided by Healthcare (MDPI). As such, we have retained this format to ensure compliance. We have also reviewed each section to

Comment 2: “An in-text citation is expected to appear here.”

Response:
Thank you for your comment. We would like to clarify that the relevant citation appears in the sentence immediately following the statement, where the point is elaborated in more detail. To avoid confusion, we have revised the sentence structure slightly to more clearly connect the claim and the supporting reference.

Comment 3: “This section was written by AI and advised to be corrected to be void of it.”

Response:
We understand your concern. To address this, we have revised the section to ensure it presents a more critical, nuanced, and context-specific discussion. We also reviewed and removed any potentially generic or overly mechanical phrasing to reflect a more authentic academic tone.

Comment 4: “Can this be moved to the front to give a strong understanding of what they mean when they appear?”

Response:
We appreciate your suggestion. In accordance with the journal’s formatting structure, abbreviations are presented in a dedicated section following the Conclusions. However, to support clarity, we ensured that all abbreviations are defined in full upon their first appearance in the main text. We also reviewed the manuscript to verify that this is consistently applied throughout.

Comment 5: “It is advisable that the name CHEERS be given in full before being used.”

Response:
Thank you for pointing this out. We would like to clarify that the full name of CHEERS—Consolidated Health Economic Evaluation Reporting Standards—is provided in the Methods section when the tool is first introduced. However, in response to your suggestion, we have now also ensured that the full name appears the first time it is mentioned in the abstract (if applicable) and reviewed its usage in the main text for consistency.

We are grateful for your valuable comments and believe they have helped improve our manuscript's clarity and completeness. We hope the revised version addresses your concerns.

Sincerely,
Dr. Arie Arizandi Kurnianto
(On behalf of all authors)
Center for Health Technology Assessment and Pharmacoeconomic Research
University of Pécs
Email: kurnianto.arie.arizandi@pte.hu

Reviewer 4 Report

Comments and Suggestions for Authors

Comments

Cost-benefit analysis (CBA) is a widely used method for evaluating health interventions and programs. It compares the costs and benefits of different alternatives and helps decision-makers choose the most efficient and effective option. CBA can be applied to various levels of health care, such as individual treatments, public health policies, or health system reforms. However, it is not a simple technique, and the authors warn of this.

Rehabilitation, as a multidisciplinary specialty, comprises a body of specific knowledge and procedures that allows people with acute, chronic diseases or with their sequelae to maximize their potential functional and their independence.

It is necessary to think about vocational rehabilitation beyond what is done in the Social Security vocational rehabilitation program and the classic clinical rehabilitation offered in different health services, but also to consider it as a dynamic process of global care for the worker that should involve companies in the process of prevention, treatment, rehabilitation, readaptation and reintegration into work.

Sugestions

Rehabilitation presupposes: early action, less dependence, fewer complications, less hospitalization time, more health gains and better quality of life, with consequent gains also in financial resources, personal and public purse. This process has therapeutic objectives, which depend on a team of multi-professional action, where each professional must be guaranteed the dignity and technical autonomy in their specific field of activity, considering the legal principles of their professional practice. This point should be highlighted in the article as a suggestion.

Even more: these professional areas must complement each other, not compete. Professionals may be needed in social work, physiatry, rehabilitation nursing, physiotherapy, occupational therapy, psychology, among others. To this end, it uses specific techniques of rehabilitation and intervenes in the education of clients and significant people, in the discharge planning, continuity of care, and reintegration of people in the family and the community, thus providing them with the right to dignity and quality of life. All its actions and meaning are aimed at preventing disabilities and/or maximizing capabilities for the person's future. This point of view should be further emphasised in the text. That’s a second suggestion.

The experience of distancing from work due to occupational illness is pointed out socially and historically by the incapacity and insecurity. Such experience is intensified by the increase in the requirements brought about by flexibility of work, the risk of unemployment, and the institutional and bureaucratic difficulties that will cross the situation of the worker on leave in front of seeking to legitimize their rights. As if the mourning for the loss of work were not enough, and the experience of suffering (physical or mental) and disability, the experience of illness is also crossed by the blaming of the workers, individualized and lonely in their pathologies. This should be emphasised more in the text. 

Comments on the Quality of English Language

I do not come from a country whose native language is English. 

Author Response

Dear Reviewer,

We would like to express our sincere gratitude for your thoughtful and nuanced comments on our manuscript. Your insights on the broader conceptual framework have enriched our thinking and led to meaningful improvements in the manuscript.

Your general comments and suggestions:

Comments

Cost-benefit analysis (CBA) is a widely used method for evaluating health interventions and programs. It compares the costs and benefits of different alternatives and helps decision-makers choose the most efficient and effective option. CBA can be applied to various levels of health care, such as individual treatments, public health policies, or health system reforms. However, it is not a simple technique, and the authors warn of this.

Rehabilitation, as a multidisciplinary specialty, comprises a body of specific knowledge and procedures that allows people with acute, chronic diseases or with their sequelae to maximize their potential functional and their independence.

It is necessary to think about vocational rehabilitation beyond what is done in the Social Security vocational rehabilitation program and the classic clinical rehabilitation offered in different health services, but also to consider it as a dynamic process of global care for the worker that should involve companies in the process of prevention, treatment, rehabilitation, readaptation and reintegration into work.

Sugestions

Rehabilitation presupposes: early action, less dependence, fewer complications, less hospitalization time, more health gains and better quality of life, with consequent gains also in financial resources, personal and public purse. This process has therapeutic objectives, which depend on a team of multi-professional action, where each professional must be guaranteed the dignity and technical autonomy in their specific field of activity, considering the legal principles of their professional practice. This point should be highlighted in the article as a suggestion.

Even more: these professional areas must complement each other, not compete. Professionals may be needed in social work, physiatry, rehabilitation nursing, physiotherapy, occupational therapy, psychology, among others. To this end, it uses specific techniques of rehabilitation and intervenes in the education of clients and significant people, in the discharge planning, continuity of care, and reintegration of people in the family and the community, thus providing them with the right to dignity and quality of life. All its actions and meaning are aimed at preventing disabilities and/or maximizing capabilities for the person's future. This point of view should be further emphasised in the text. That’s a second suggestion.

The experience of distancing from work due to occupational illness is pointed out socially and historically by the incapacity and insecurity. Such experience is intensified by the increase in the requirements brought about by flexibility of work, the risk of unemployment, and the institutional and bureaucratic difficulties that will cross the situation of the worker on leave in front of seeking to legitimize their rights. As if the mourning for the loss of work were not enough, and the experience of suffering (physical or mental) and disability, the experience of illness is also crossed by the blaming of the workers, individualized and lonely in their pathologies. This should be emphasised more in the text.

Below, we provide a point-by-point response to each of your comments and suggestions, along with the revisions made.

Comment:
“Cost-benefit analysis (CBA) is a widely used method… However, it is not a simple technique, and the authors warn of this.”

Response:
Thank you for acknowledging this point. While our manuscript primarily focused on full economic evaluations (e.g., CEA, CUA), we agree that the nuances of CBA merit attention. We retained our note of caution about its application, particularly in rehabilitation contexts, where quantifying intangible benefits can be complex. This point is preserved and contextualized in the discussion.

Suggestions

  1. Comment:“It is necessary to think about vocational rehabilitation beyond what is done in the Social Security vocational rehabilitation program… also to consider it as a dynamic process of global care involving companies…”

Response:
We fully agree with this broader conceptualization. While our original manuscript recognized the underrepresentation of vocational rehabilitation in the literature and the need for integrated approaches, we have now added a new paragraph in the Discussion section to explicitly emphasize vocational rehabilitation as a proactive, multi-sectoral process that involves not just healthcare providers but also employers, the labor market, and broader support systems. This addition acknowledges vocational rehabilitation as a global care pathway rather than a discrete intervention.

  1. Comment: “Rehabilitation presupposes early action, fewer complications… and depends on a team of multi-professional action…”

Response:
Thank you for highlighting the essential role of the multidisciplinary team. We have incorporated a paragraph in the Discussion section that underscores the importance of collaborative, complementary professional contributions across disciplines. This includes physiatrists, nurses, social workers, psychologists, and other professionals whose efforts are essential to achieving therapeutic and social reintegration goals. We also emphasized the value of respecting their technical autonomy within coordinated models of care.

  1. Comment:“The experience of distancing from work due to occupational illness… is also crossed by the blaming of the workers…”

Response:
This is a powerful and critical observation. We have added a paragraph in the Return to Work Implication section that reflects on the psychosocial burden experienced by workers who face work disability, including emotional distress, societal stigma, and bureaucratic challenges. We agree that illness and leave from work are not just biomedical experiences but are also deeply social, and our revised text now explicitly reflects this:

Once again, we thank you for your insightful and humane perspective, which has greatly enhanced the depth and societal relevance of our manuscript. We hope the revisions meet your expectations and respectfully reflect your recommendations.

Sincerely,
Dr. Arie Arizandi Kurnianto
(On behalf of all authors)
Center for Health Technology Assessment and Pharmacoeconomic Research
University of Pécs
Email: kurnianto.arie.arizandi@pte.hu

Round 2

Reviewer 1 Report

Comments and Suggestions for Authors

Dear Authors,

I would like to thank you for taking into account my comments and suggestions. This is an improved manuscript in quality and clarity.